# Impact of Feeding Syngenta Enogen^®^ Feed Corn Compared to Control Corn in Different Diet Scenarios to Finishing Beef Cattle

**DOI:** 10.3390/ani11102940

**Published:** 2021-10-11

**Authors:** Stacia M. Volk, Hannah C. Wilson, Kathryn J. Hanford, James C. MacDonald, Galen E. Erickson

**Affiliations:** 1Department of Animal Science, University of Nebraska, Lincoln, NE 68583-0908, USA; stacia.hopfauf@huskers.unl.edu (S.M.V.); hannah.wilson2@usda.gov (H.C.W.); jmacdonald2@unl.edu (J.C.M.); 2Department of Statistics, University of Nebraska, Lincoln, NE 68583-0908, USA; kathy.hanford@unl.edu

**Keywords:** α-amylase, beef cattle, byproducts, corn hybrid, corn trait, starch

## Abstract

**Simple Summary:**

A pooled statistical analysis of seven experiments and 200 pen observations was performed to determine the impact of feeding Enogen^®^ corn compared to conventional corn grain in beef cattle finishing diets. When the corn was compared as dry-rolled corn in diets with lower inclusion rates of distiller grains plus solubles (less than 20% of the diet), feeding Enogen^®^ corn improved the efficiency of beef production. That response was a 4.8% improvement due to feeding Enogen^®^ in a corn-based diet without distiller grains but was 1.8% in diets with 18 to 20% distiller grains. Feeding Enogen^®^ corn improved the efficiency by 4.5% in diets with another common byproduct, Sweet Bran^®^. Cattle performance was similar for Enogen^®^ and conventional hybrids when processed and fed as high-moisture corn. Feeding Enogen^®^ corn improves the gain efficiency of finishing cattle compared with conventional corn when fed as dry-rolled corn in diets with less than 20% distillers or diets that include Sweet Bran^®^ feeds.

**Abstract:**

The objective of this pooled statistical analysis was to evaluate Syngenta Enogen^®^ Feed Corn (EFC) versus conventional corn (CON) when fed as either dry-rolled corn (DRC) or high-moisture corn (HMC) for effects on finishing beef cattle performance and carcass characteristics. Corns were evaluated in diets with byproduct inclusion rates of 0, 15, 18, 20, and 30% distiller grains or 25 and 35% Sweet Bran^®^ (a commercial corn gluten feed product). Seven trials (*n* = 1856) consisting of 200 pen means comparing 26 diet treatments were analyzed using regression in a pooled analysis. When EFC was processed as DRC, the gain efficiency (G:F) improved compared with CON, but the response to feeding EFC decreased from a 4.8% improvement to no improvement compared to CON as distiller grains increased from 0 to 30%, but was significantly improved due to feeding EFC in diets with 0 to 18% distiller grains. Feeding cattle EFC as DRC increased the average daily gain (ADG) and G:F by 4.5% compared with CON corn in diets containing Sweet Bran^®^. No improvements in animal performance were observed when cattle were fed EFC compared to CON when processed as HMC in any situation. Feeding Enogen^®^ corn improved the gain efficiency of finishing cattle compared with conventional corn when processed as dry-rolled corn and fed in diets with less than 20% distillers or diets that include Sweet Bran^®^.

## 1. Introduction

Corn is the most widely fed grain source in finishing beef cattle diets in the United States [1] and starch is the main energy component in corn. Therefore, improving starch digestion could improve cattle performance. Different corn processing methods have been used to increase starch digestion in finishing cattle. Common corn processing methods include ensiling as high-moisture corn (HMC), rolling as dry-rolled corn (DRC), and flaking to produce steam-flaked corn. Corn processing is used to increase the digestion of starch [2,3] and improve gain efficiency. Starch digestion mainly occurs in the rumen; however, increased starch digestion in the rumen increases the potential for digestive disturbances [4]. Therefore, new research has been focused on ways to improve starch digestion post-ruminally, which may improve performance without increasing digestive disturbances in cattle fed high-concentrate diets.

Supplementing exogenous alpha amylase in finishing cattle diets has been shown to improve cattle performance [5]. Burroughs et al. [6] observed increases in BW and improved gain efficiency when exogenous amylase was supplemented in high-forage feedlot diets. In high-grain diets, the response to adding amylase may depend on the corn processing method. Zerby et al. [7] observed a 7.2% improvement in gain efficiency when finishing steers were fed supplemental amylase in dry whole corn diets, but not in HMC-based diets.

Syngenta Enogen^®^ feed corn (EFC; Syngenta Seeds, LLC, Downers Grove, IL, USA) is a genetically enhanced corn hybrid that contains a thermotolerant alpha-amylase enzyme trait that is activated at increased temperatures intended for the dry-milling ethanol process. This trait in the corn eliminates the need for exogenous amylase to convert starch to sugar before ethanol fermentation, thereby reducing the cost of adding exogenous amylase. Feeding EFC to finishing cattle may improve performance by increasing post-ruminal starch digestion. An increase in post-ruminal and total tract starch digestion was observed when EFC was processed as dry-rolled corn (DRC) in comparison with cattle fed corn without the alpha-amylase enzyme trait [8]. When EFC was processed as DRC and fed with wet distiller grains (WDGS), a greater final BW and ADG were observed with a 5.7% improvement in G:F [8]. In the same study, when EFC was processed as DRC and fed with 0 WDGS, ADG and G:F were not significantly different between EFC and corn with the amylase trait, but cattle had a numerically greater ADG and a 3% improvement in G:F compared with those fed CON corn with no WDGS (*p* = 0.17) [9]. Cattle fed EFC (processed as DRC) with 18% WDGS had no significant differences for carcass-adjusted final BW, DMI, ADG, and G:F (*p* ≥ 0.17) compared to steers fed a CON corn with 18% WDGS [8]. Previous research has shown a greater ADG and a 10.1% improvement in G:F when EFC was processed as DRC and fed in combination with Sweet Bran^®^ [10]. However, no significant differences were observed when EFC was processed as HMC in comparison to cattle fed corn without the alpha-amylase enzyme trait [9]. Cattle fed a corn without the amylase trait (processed as HMC) had a greater final BW and ADG (*p* = 0.03) with a similar G:F (*p* = 0.16) compared with cattle fed EFC [9]. Horton et al. [11] fed EFC processed as steam-flaked corn without distiller grains to finishing heifers. Cattle fed EFC had a greater final BW and ADG with a 4.8% improvement in G:F compared with a CON diet. Heifers fed EFC had a heavier HCW compared with CON corn. In contrast, Baker et al. [12] fed cattle EFC as steam-flaked corn and observed no differences in final BW, DMI, and ADG with a decreased G:F compared with CON diets.

Overall, the response of EFC has been variable across studies depending on the corn processing method and byproduct utilized. In addition, studies conducted at the University of Nebraska-Lincoln (UNL) have shown small numerical improvements that were often not significant statistically. Therefore, the objective was to evaluate the impact of EFC processed as DRC or HMC in combination with varying inclusion rates of distiller grains or Sweet Bran^®^ on finishing cattle performance and carcass characteristics.

## 2. Materials and Methods

Data were collected from seven experiments conducted from 2012 to 2018 at the Eastern Nebraska Research and Extension Center near Mead, NE and the Panhandle Research and Extension Center near Scottsbluff, NE. The data included 200 pen means (*n* = 1856 heads) fed 26 different treatment diets of EFC or CON corn fed as DRC or HMC, and varying inclusion rates of distiller grains plus solubles (0, 15, 18, 20, or 30% of diet DM) or a commercial corn gluten feed (Sweet Bran^®^, Cargill Inc., Blair, NE, USA) at 25 or 35% (Table 1). Experiment 1 [8] utilized 240 heads with 10 heads/pen and six replications per treatment. The four treatments utilized consisted of (1) CON fed as DRC with 15% distiller grains plus solubles, (2) EFC fed as DRC with 15% distiller grains plus solubles, (3) CON fed as DRC with 25% Sweet Bran^®^, and (4) EFC fed as DRC with 25% Sweet Bran^®^. A fifth treatment was omitted from the pooled analysis (CON:EFC fed as a 50:50 blend as DRC) as our objective did not include pooling corn blends. Experiment 2 [8] utilized 120 heads with 10 heads/pen and six replications per treatment. The two treatments utilized in this study were (1) CON fed as DRC with 15% distiller grains plus solubles and (2) EFC fed as DRC with 15% distiller grains plus solubles. Treatments were similar to those used in Experiment 1 but at a different location and time. Two treatments that were not included in our pooled data were (1) CON:EFC as a 50:50 blend with 15% distiller grains plus solubles and (2) CON fed as dry-rolled corn + an enzyme. Experiment 3 [12] utilized 384 heads with 8 heads/pen and six replications per treatment. The eight treatments utilized in this study were (1) CON fed as DRC with 18% distiller grains plus solubles, (2) EFC fed as DRC with 18% distiller grains plus solubles, (3) CON fed as DRC with 35% Sweet Bran^®^, (4) EFC fed as DRC with 35% Sweet Bran^®^, (5) CON fed as HMC with 18% distiller grains plus solubles, (6) EFC fed as HMC with 18% distiller grains plus solubles, (7) CON fed as HMC with 35% Sweet Bran^®^, and (8) EFC fed as HMC with 35% Sweet Bran^®^. Experiment 4 and 5 [12] both used 300 heads per trial with 10 heads/pen and 30 total replications per treatment. Treatments consisted of (1) CON fed as DRC with 18% distiller grains plus solubles or (2) EFC fed as DRC with 18% distiller grains. Experiment 6 [9] used 192 heads with 8 head per pen and six replications per treatment. Treatments consisted of (1) CON fed as DRC with 20% distiller grains plus solubles, (2) EFC fed as DRC with 20% distiller grains plus solubles, (3) CON fed as HMC with 20% distiller grains plus solubles, and (4) EFC fed as HMC with 20% distiller grains plus solubles. Two treatments were omitted from the pooled analysis, consisting of (1) HMC:DRC fed with 20% distiller grains plus solubles and (2) EFC blended with CON corn fed with 20% distiller grains plus solubles. Experiment 7 [9] used 320 heads with 10 heads per pen and eight replications per treatment. Treatments consisted of (1) CON fed as DRC with 0 distiller grains plus solubles, (2) EFC fed as DRC with 0 distiller grains plus solubles, (3) CON fed as DRC with 30% distiller grains plus solubles, and (4) EFC fed as DRC with 30% distiller grains plus solubles. Two treatments were removed from the pooled analysis: (1) EFC fed as DRC with 15% distiller grains plus solubles and (2) EFC fed as DRC with 45% distiller grains plus solubles, as no CON treatment was used to compare both treatments at those inclusion rates of distiller grains plus solubles. In studies that fed HMC, corn was harvested at a DM from 68 to 72% DM and stored for 2 to 8 months prior to feeding. Corn was rolled prior to ensiling in silo bags.

Hot carcass weights were recorded on the day of harvest for all experiments and carcass-adjusted final BW was calculated from a common 63.0% dress. Carcass-adjusted final BW was used to determine ADG and gain efficiency (G:F). Carcass characteristics included marbling score, *Longissimus* muscle (LM) area, and 12th rib fat, which were recorded after a 48 h chill. In all experiments, monensin (Rumensin, Elanco Animal Health) and tylosin (Tylan, Elanco Animal Health) were included in the diet and cattle were implanted on d1 and re-implanted during the different experiments.

All analyses were performed using SAS 9.4 PROC GLIMMIX (SAS, Cary, NC, USA) with pen as the experimental unit. In order to account for possible differences due to random trial effects, data were analyzed using the meta-analysis approach [13]. A similar initial model was used to analyze all traits. The initial model included the fixed effects of corn type (TYPE; CON vs. EFC), processing method (PROCESSING; DRC vs. HMC), linear and quadratic effects of distiller grains plus solubles inclusion rate, effects of Sweet Bran^®^ inclusion rate, the TYPE × PROCESSING interaction effect, all possible linear DG × TYPE and DG × PROCESSING interactions, all possible SB × TYPE and SB × PROCESSING interactions and the random effects of trial (TRIAL), weight block nested within the trial, and residual error. Non-significant interactions and quadratic terms (*p* > 0.05) were dropped to produce the final model for each trait. A normal distribution was assumed for all traits measured. Significance was determined at *p* ≤ 0.05. In order to estimate predicted values for the treatments and their differences, estimate statements were constructed at the various levels of inclusion from the original studies (i.e., DG inclusion rates of 0, 15, 18, 20, and 30% and 25 or 35% SB). Therefore, all data means presented were modeled from the complete datasets and are LSMeans predicted values only. To account for the multiple comparisons of TYPE in DRC, a Bonferroni adjustment was used.

## 3. Results

### 3.1. Enogen^®^ Corn Fed with Distiller Grains

Table 2 presents the predicted values for the linear interaction between the corn hybrids when fed as DRC and distiller grains plus solubles. No difference was detected for final body weight between cattle fed EFC compared to CON diets. When cattle were fed EFC as DRC with 15% distiller grains plus solubles, DMI (*p* = 0.10) tended to be lower for cattle fed EFC compared with the CON diets. However, DMI was not different between EFC or CON when modeled for all other distiller grains inclusion rates. Cattle fed EFC as DRC with a distiller grains plus solubles inclusion rate from 0 to 30% DG did not differ in ADG (*p* > 0.83) compared to cattle fed the CON corn. As a result of a numerically lower DMI and a similar, but numerically greater ADG, the gain efficiency was impacted by the corn hybrids. A linear interaction was observed between the corn hybrids and the distiller grains plus solubles inclusion rate. The impact on G:F due to the corn hybrid appears to be dependent on the distiller grains plus solubles inclusion rate. Cattle fed EFC as DRC with a 0%, 15%, and 18% distiller grains inclusion rate had a 4.8%, 2.3%, and 1.9% improvement in G:F (*p* ≤ 0.03) compared with CON diets, respectively (Figure 1). Cattle fed EFC with 20% distillers were numerically (*p* = 0.12) more efficient than cattle fed CON with 20% distillers. When cattle were fed either hybrid with 30% distiller grains, the G:F was identical. These data suggest that the response to EFC for improving gain efficiency above CON hybrids is dependent on the inclusion rate of distiller grains plus solubles. As more distillers are fed (>20% inclusion rate), no difference is expected between CON and EFC when fed as DRC. In both cases, feeding more distiller grains plus solubles (in these studies as either wet or modified distiller grains plus solubles) increased G:F. Additionally, fecal starch was reduced as the distiller grains inclusion rate increased for both EFC and CON (Experiment 7) but can be influenced by diet DM digestibility. Fecal starch was consistently lower in animals fed EFC, with 19% lower fecal starch for EFC compared with CON when fed with 0 distiller grains (*p* < 0.01) and 23% lower when fed with 30% distiller grains (*p* = 0.02).

No differences for HCW (*p* ≥ 0.49) were observed between steers fed EFC and CON diets across all inclusion rates of distiller grains plus solubles (Table 2). A significant difference was observed for 12th rib fat thickness (*p* ≤ 0.01), with cattle fed EFC as DRC with 15%, 18%, and 20% distiller grains plus solubles having a greater 12th rib fat thickness compared with the CON diet, but 12th rib fat thickness was not statistically different for EFC compared to the control when fed with 0 or 30% distiller grains plus solubles. No differences were observed for LM area (*p* ≥ 0.83) when cattle were fed either EFC or CON corn with a distiller grains plus solubles inclusion rate from 0 to 30% of diet DM. Cattle fed EFC with 0 to 30% distiller grains plus solubles had a similar marbling score (*p* ≥ 0.22) as cattle fed CON corn at those same distiller grain inclusion rates.

### 3.2. Enogen^®^ Corn Fed with Sweet Bran^®^

Two experiments were included in the pooled analysis that evaluated the effect of the corn hybrids fed as DRC with 25% and 35% Sweet Bran^®^ (Table 3). No differences were observed between hybrid treatments for DMI, LM area, or fat thickness. Cattle fed EFC when processed as DRC and fed with either 25 or 35% Sweet Bran^®^ had a greater ADG and G:F (*p* < 0.05). The increase in G:F was a 4.8% improvement for cattle fed EFC compared with CON.

### 3.3. Enogen^®^ Corn Fed as High-Moisture Corn

Two experiments were included in the pooled analysis to evaluate the effect of the corn hybrids when processed as HMC and fed with 18 or 20% distiller grains plus solubles (Table 4). No significant differences were observed between EFC and CON corn when processed as HMC and fed with distiller grains plus solubles for DMI and G:F (*p* ≥ 0.13). However, when cattle were fed EFC as HMC with distiller grains plus solubles, the final BW and ADG were decreased (*p* < 0.05) compared with CON corn fed as HMC. No differences were observed for 12th rib fat thickness, LM area, and marbling score (*p* ≥ 0.32) between EFC and CON corn processed as HMC with distiller grains plus solubles. These data suggest no benefit, and perhaps a detrimental gain response of feeding EFC as HMC in diets with distiller grains.

One experiment was included in the pooled analysis to evaluate the effect of the corn hybrids (HMC) with 35% Sweet Bran^®^ (Table 5). Cattle fed EFC as HMC with 35% Sweet Bran^®^ had a tendency for a lighter final BW (*p* = 0.08) and a reduced ADG (*p* = 0.08), but DMI and G:F were not significantly different (*p* > 0.27) compared to cattle fed CON diets. A tendency for a lighter HCW (*p* = 0.08) was observed for cattle fed EFC as HMC with 35% Sweet Bran^®^ compared with CON diets. No differences were observed for 12th rib fat thickness, marbling score, and LM area (*p* ≥ 0.18) between CON corn and EFC diets fed as HMC.

## 4. Discussion

Cattle appear to respond favorably to feeding EFC as DRC if diets contain less than 20% distiller grains plus solubles. The inclusion rate and source (wet, dried, modified) of distiller grains vary across time and regions. In general, the inclusion rate averaged 20% of diet DM in the Midwest and 26% in the Northern Plains [14], but is likely lower (10 to 15%) in the southern plains. The improved G:F observed for EFC appears to be a typical energy response for cattle fed concentrate diets whereby DMI was lower while ADG was similar for cattle fed EFC as compared with CON. It is unclear why the response to feeding EFC is influenced by the distiller grains inclusion rate. One explanation is simply due to starch or enzyme present in the diet, as the response was better as more corn was included (i.e., in diets with less distiller grains plus solubles). However, improvements in gain efficiency were observed when feeding EFC as DRC in diets with 25 or 35% Sweet Bran^®^ compared with CON corn and were due primarily to increased gain. The improved G:F for EFC fed as DRC when Sweet Bran^®^ was included at 25 or 35% suggests that starch intake does not influence the response to EFC, unlike what was observed for EFC fed as DRC with greater inclusion rates of distiller grains. Schoonmaker et al. [15] fed a corn hybrid containing an amylase enzyme at 0%, 10%, and 20% of the diet and processed as DRC with 20% DG. The remainder of the corn mix was conventional corn in their diets. The final body weight, DMI, and ADG were similar among corn treatments, suggesting no impact of feeding Enogen^®^ corn at 20% of the diet or less (or as a proportion). In their study, gain efficiency was not statistically different (*p* = 0.28); however, a calculated 5.2% and 4.6% improvement was observed for the 0 treatment compared with the 10% and 20% treatments, respectively. Johnson et al. [16] compared EFC to conventional corn when fed as DRC to growing calves along with 30% distiller grains. A greater final BW and ADG were observed with a 2.4% improvement in G:F compared with CON diets for these growing diets with 38.5% corn in the diet.

When evaluating the individual studies that comprised this pooled analysis, there was less power/repetition, resulting in no statistical differences detected for DMI and ADG when comparing EFC to CON corn fed with 0 [9], 18% [12], or 30% [9] distiller grains plus solubles. Improvements of 3.3% and 1% in G:F were observed when cattle were fed EFC with 0 and 18% DG, respectively, but were not significant (*p* = 0.17) [9,12]. Similar intakes with a greater ADG and a 5.4% improvement in G:F were observed for EFC fed with 15% DG compared with conventional corn [8]. Brinton [9] observed a lower DMI with a similar gain and a 2.9% improvement in G:F (*p* ≥ 0.17) for cattle fed CON corn as DRC with 20% DG compared with cattle fed EFC. The results from the pooled analysis summarize the individual studies and allow for greater confidence in the predicted response.

If high inclusion rates of distiller grains plus solubles are fed, the impact of feeding EFC does not change cattle performance relative to conventional corn. As for why cattle respond differently to feeding EFC compared with conventional corn when diets contain more distiller grains plus solubles, that response is interesting and actually similar to previous work on corn processing. Steam-flaking corn improves intestinal digestion [17] and improves gain efficiency by 12% [2] compared with DRC when fed in diets without byproducts. In diets with 30% distiller grains, steam-flaking did not improve G:F compared to DRC [18], and adding distiller grains to the diet in place of steam-flaked corn results in a much smaller response to the distiller grains [3,19,20]. Similar to the response observed in this pooled analysis, the response to steam-flaking corn in diets with Sweet Bran^®^ shows a 12% or greater improvement in G:F compared with DRC [21,22]. So, while the results suggest a diminishing response to feeding EFC compared with CON hybrids as the distiller grains plus solubles inclusion rate increases, a similar issue has been documented when corn is processed as steam-flaked corn.

Feeding EFC improves starch digestion in the small intestine in DRC-based diets [8] and other research has evaluated the effect of either enzyme additions or additives that provide an amylase enzyme, such as *Aspergillus oryzae*, on finishing performance. When finishing lambs were fed a ground dry corn diet, the gain efficiency was improved with *Aspergillus oryzae* supplementation [7]. In the same paper, cattle fed either dry-rolled or high-moisture corn did not respond to supplementation. Conversely, DiLorenzo et al. [23] fed amylase to cattle in diets consisting of either steam-flaked or dry-rolled corn. While the authors concluded that enzyme addition did not impact performance and no interaction with the corn processing method, cattle fed dry-rolled corn with enzyme were 9.4% more efficient than cattle fed dry-rolled corn without enzyme, suggesting an effect; however, the study lacked sufficient statistical power to detect the difference. Tracarico et al. [5] presented three experiments evaluating amylase added to grain diets. The authors concluded no statistical difference due to enzyme addition in one of their experiments evaluating the response to cracked corn diets. However, the gain efficiency was improved by 9.6% at the lower addition rate of enzyme and 11.6% at the higher inclusion rate of enzyme, suggesting a biological response.

In the pooled analysis, no response was observed for feeding EFC as high-moisture corn compared to a CON hybrid. Data on adding amylase to grain-based diets based on high-moisture corn are conflicting in the literature. Tricarico et al. [5] observed an increased ADG (*p* = 0.04) due to an increased DMI, resulting in no difference in G:F when cattle were fed HMC with supplemental alpha amylase. Cattle fed HMC with amylase had a heavier HCW and a greater 12th rib fat thickness (*p* ≤ 0.05) compared with diets without supplemental amylase. Similar to the pooled analysis from our results, Zerby et al. [7] found no differences in final BW, DMI, and ADG for cattle fed HMC supplemented with amylase without distiller byproducts. Cattle fed HMC with amylase had a G:F that was not significantly different to cattle fed HMC without enzyme (*p* = 0.61). No differences were observed for HCW, 12th rib fat thickness, LM area, and marbling for steers fed HMC-based diets supplemented with an exogenous amylase enzyme.

Regardless of whether cattle were fed EFC or CON corn, ADG and G:F improved as distiller grains plus solubles were added to the diet (Table 2), just at different rates (Figure 1). Research clearly shows that in diets with dry-rolled corn replaced with wet or modified distiller grains plus solubles, gain and gain efficiency are dramatically improved [24,25].

## 5. Conclusions

Previous research has observed small improvements or no improvements in animal performance when cattle were fed EFC processed as DRC with distiller grains. The pooled analysis would suggest that, when EFC is processed as DRC and fed with 0 to 20% distiller grains, an improvement in G:F would be expected compared with conventional corn. However, as the distiller grains inclusion rate increases in the diet, the improvement due to feeding EFC compared with conventional corn tends to decrease from 4.8% to 0 for gain efficiency. In addition, a 4.8% improvement in gain efficiency was observed when EFC was fed as DRC compared with conventional corn in diets with Sweet Bran^®^. These data suggest no improvements in animal performance when cattle are fed EFC as HMC when compared to a conventional corn, at least based on how HMC was processed in these studies. It appears that processing Enogen^®^ corn as dry-rolled corn will lead to slight improvements in gain efficiency in most diet scenarios, but improvements are not expected when feeding Enogen^®^ corn when processed as high-moisture corn.

## Figures and Tables

**Figure 1 animals-11-02940-f001:**
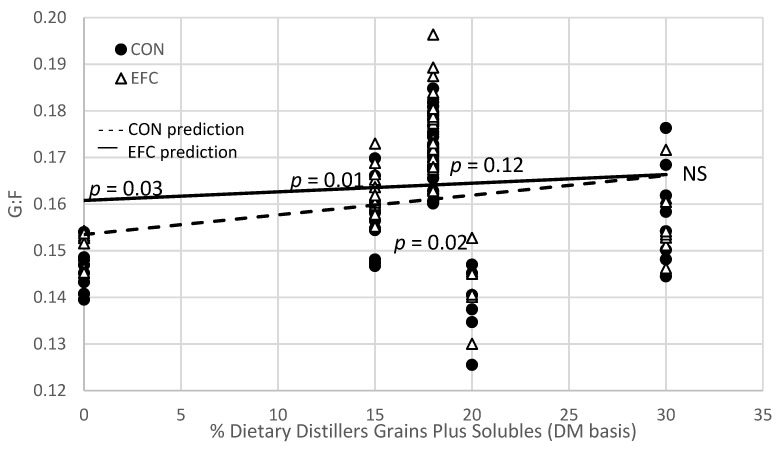
Linear interaction between corn hybrid (CON vs. EFC) fed as dry-rolled corn and distiller grains plus solubles inclusion rate for G:F.

**Table 1 animals-11-02940-t001:** Pens (*n*) per corn hybrid ^1^ (EFC and CON) ^1^ fed as dry-rolled corn or high-moisture corn in diets with Sweet Bran^®^ or distiller grains plus solubles included in the pooled analysis. Within the hybrid, 100 pens of data were compiled.

Study	Grain Processing ^3^	Distiller Grains Plus Solubles Inclusion Rate					Sweet Bran^®^ Inclusion ^2^	
		0	15	18	20	30	25	35
1	DRC	-	6	-	-	-	6	-
2	DRC	-	6	-	-	-	-	-
3	DRC	-	-	6	-	-	-	6
4	DRC	-	-	15	-	-	-	-
5	DRC	-	-	15	-	-	-	-
6	DRC	-	-	-	6	-	-	-
7	DRC	8	-	-	-	8	-	-
Pens/hybrid	-	8	12	36	6	8	6	6
3	HMC	-	-	6	-	-	-	6
6	HMC	-	-	-	6	-	-	-
Pens/hybrid	-	0	0	6	6	0	0	6

^1^ Hybrids consisted of Control corn (CON) or Enogen^®^ Feed Corn (EFC); ^2^ Different inclusion rates of either distiller grains plus solubles or a branded corn gluten feed (Sweet Bran^®^) were fed in different studies; ^3^ Corn was processed as either dry-rolled corn (DRC) or high-moisture corn (HMC).

**Table 2 animals-11-02940-t002:** Predicted performance outcomes on the effect of the corn hybrids fed as dry-rolled corn in combination with varying inclusion rates of distiller grains on predicted least-square means when finishing cattle performance and carcass characteristics are modeled.

	0 DG ^1^		15 DG ^1^		18 DG ^1^		20 DG ^1^		30 DG ^1^		SED ^2^	*p*-Value ^3^				
	CON ^4^	EFC ^5^	CON ^4^	EFC ^5^	CON ^4^	EFC ^5^	CON ^4^	EFC ^5^	CON ^4^	EFC ^5^		0 DG	15 DG	18 DG	20 DG	30 DG
Performance ^6^																
Initial BW, kg	330	330	330	330	330	330	330	330	330	330	0.2–0.5	1.0	1.0	1.0	1.0	1.0
Final BW ^7^, kg	604	609	617	620	619	622	621	624	630	631	2.0–5.0	1.0	0.49	0.65	0.96	1.0
DMI, kg/d	10.8	10.5	10.9	10.7	10.9	10.8	10.9	10.8	10.9	10.9	0.06–0.014 0.14	0.25	0.10	0.29	0.70	1.0
ADG, kg	1.65	1.68	1.73	1.75	1.75	1.76	1.76	1.77	1.81	1.82	0.013–0.131	1.0	0.83	1.0	1.0	1.0
G:F	0.1535	0.1608	0.1598	0.1635	0.1611	0.1641	0.1619	0.1645	0.1661	0.1663	0.00158–0.00256	0.03	0.01	0.02	0.12	1.0
Carcass Characteristics ^8^																
HCW, kg	380	384	389	391	390	392	391	393	397	398	1.3–3.2	1.0	0.49	0.65	0.96	1.0
LM area, cm^2^	86.2	86.3	86.3	85.8	86.3	85.7	86.3	85.6	86.4	85.2	0.50–1.23	1.0	1.0	0.93	0.83	1.0
Fat depth, cm	1.23	1.28	1.39	1.46	1.42	1.50	1.45	1.52	1.56	1.64	0.023–0.057	1.0	0.02	0.01	0.02	0.34
Marbling ^9^	493	516	500	509	501	508	502	507	506	502	7.5–18.3	1.0	1.0	1.0	1.0	1.0

^1^ DG, distiller grains inclusion rate in the diet (% of diet DM); ^2^ SED, standard error of the difference between Enogen^®^ Feed Corn and Control corn within 0 DG (0DG), 15% DG (15DG), 18% DG (18DG), 20% DG (20DG), and 30% DG (30DG) diets; ^3^ *p*-Value with Bonferonni adjustment for multiple comparisons representing the difference between Enogen^®^ Feed Corn and Control corn within 0 DG (0DG), 15% DG (15DG), 18% DG (18DG), 20% DG (20DG), and 30% DG (30DG) diets; ^4^ CON, Control corn; ^5^ EFC, Enogen^®^ Feed Corn; ^6^ BW, body weight; DMI, dry matter intake; ADG, average daily gain, G:F, ADG/DMI; ^7^ Calculated from HCW adjusted to a common 63.0% dress; ^8^ HCW, hot carcass weight; LM, *longissimus* muscle; ^9^ Marbling score: 400 = Small^00^, 500 = Modest^00^.

**Table 3 animals-11-02940-t003:** Effect of the corn hybrids (EFC and CON) fed as dry-rolled corn in diets with Sweet Bran^®^ on predicted least-square means when finishing cattle performance and carcass characteristics are modeled.

	Treatments ^1^				SED	*p*-Value ^2^	
	25% Sweet Bran^®^		35% Sweet Bran^®^				
	CON	EFC	CON	EFC		25% SB	35% SB
Performance							
Initial BW, kg	330	330	330	330	0.4–0.5	1.0	1.0
Final BW ^3^, kg	618	629	624	636	4.1–5.7	0.03	0.06
DMI, kg/d	11.0	10.9	11.1	11.1	0.12–0.16	1.0	1.0
ADG, kg	1.73	1.80	1.77	1.85	0.026–0.036	0.02	0.05
G:F	0.1580	0.1656	0.1598	0.1675	0.00214–0.00297	<0.01	0.02
Carcass Characteristics							
HCW, kg	389	396	393	401	2.6–3.0	0.03	0.06
LM area, cm^2^	86.1	87.1	86.1	87.4	1.01–1.39	0.70	0.70
Fat depth, cm	1.39	1.40	1.45	1.45	0.047–0.064	1.0	1.0
Marbling ^4^	496	528	498	534	15.1–20.7	0.07	0.18

^1^ Treatments consisted of Control corn (CON) or Enogen^®^ Feed Corn (EFC) with 25% or 35% Sweet Bran^®^ (DM basis); ^2^ *p*-value with Bonferonni adjustment for multiple comparisons representing the difference between EFC and CON within 25% Sweet Bran^®^ (25% SB) and 35% Sweet Bran^®^ (35% SB) diets; ^3^ Calculated from HCW adjusted to a common 63.0% dress; ^4^ Marbling score: 400 = Small^00^, 500 = Modest^0^.

**Table 4 animals-11-02940-t004:** Main effect of the corn hybrids (EFC vs. CON) fed as high-moisture corn with 18 or 20% distiller grains on predicted least-square means when finishing cattle performance and carcass characteristics are modeled.

	Treatments ^1^		SED	*p*-Value
	CON	EFC		
Performance				
Initial BW, kg	329	329	0.67	0.84
Final BW ^2^, kg	633	619	6.9	0.05
DMI, kg/d	10.1	9.9	0.19	0.26
ADG, kg	1.83	1.73	0.043	0.03
G:F	0.1786	0.1732	0.00359	0.13
Carcass Characteristics				
HCW, kg	399	390	4.4	0.05
LM area, cm^2^	90.2	88.5	1.69	0.32
Fat depth, cm	1.50	1.50	0.079	0.99
Marbling ^3^	520	520	25.3	0.98

^1^ Treatments consisted of Control corn (CON) or Enogen^®^ Feed Corn (EFC) with distiller grains plus solubles; ^2^ Calculated from HCW adjusted to a common 63.0% dress; ^3^ Marbling score: 400 = Small^00^, 500 = Modest^0^.

**Table 5 animals-11-02940-t005:** Main effect of the corn hybrids (CON vs. EFC) when fed as high-moisture corn in diets with 35% Sweet Bran^®^ on predicted least-square means when finishing cattle performance and carcass characteristics are modeled.

	Treatments ^1^		SED	*p*-Value
	CON	EFC		
Performance				
Initial BW, kg	330	329	0.7	1.0
Final BW ^2^, kg	634	621	6.9	0.08
DMI, kg/d	10.5	10.4	0.20	0.34
ADG, kg	1.83	1.76	0.043	0.08
G:F	0.1754	0.1714	0.00359	0.27
Carcass Characteristics				
HCW, kg	399	391	4.4	0.08
LM area, cm²	86.3	84.5	1.69	0.28
Fat depth, cm	1.47	1.58	0.079	0.18
Marbling ^3^	534	514	25	0.44

^1^ Treatments consisted of Control corn (CON) or Enogen^®^ Feed Corn (EFC) with 35% Sweet Bran^®^ (SB); ^2^ Calculated from HCW adjusted to a common 63.0% dress; ^3^ Marbling score: 400 = Small^00^, 500 = Modest^0^.

## Data Availability

Data sharing not applicable.

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
