# Peer review of "Impact of Feeding Syngenta Enogen® Feed Corn Compared to Control Corn in Different Diet Scenarios to Finishing Beef Cattle"

_animals, 2021, doi:10.3390/ani11102940_

Round 1
Reviewer 1 Report
To Authors;
The manuscript is of potential interest to the industry, but I’m not sure it adds much to our current knowledge base. It “summarizes multiple published studies conducted over several years at a single location. The studies have no common treatment(s) with which to equalize/correct values (although initial weights [and other data??] seem to have been equalized by some undefined method). For most treatment comparisons, results of only one study are used (so why the “pooled” analysis?). How do the authors know the effects are truly a treatment effect or a time, or study effect? The statistical methods need to be evaluated by someone familiar with meta-analysis.
In my opinion the use of a commercial name (Sweet Bran) throughout the manuscript is inappropriate. Although the commercial name and source should be mentioned in the M&M, the term “commercial corn gluten feed – or maybe CCGF” would be more appropriate for a scientific publication.
Authors frequently discuss nonsignificant differences as if they are significant. These parts need to be reworded.
The composition of the distillers grains and corn gluten feed and diets used should be mentioned in the text or a table.
In some places authors use “feed efficiency” (feed : gain I assume) and in other use gain : feed. Be consistent throughout and use only one term.
The manuscript needs some rewriting – there are a number of incomplete sentences, unneeded repetition, and complex/wordy sentences that are difficult to understand. In at least one case the references provided in the Introduction , M&M and Discussion are not the correct manuscript. See specific comments.
Line Comment
23 “n=1856 cattle) … regression.” 20 treatments? The M&M only mention 18.
24 “decreased”
25 authors discuss differences that are not statistically different as if they are different – in my opinion discussing “0.1% better” is inappropriate.
25 “increased”
29 “improved”
37 provide a reference.
61-63 I believe reference is incorrect – Enogen was not fed in reference 8. Double check all references.
73 be consistent with acronyms.
M&M recommend all the studies with treatments, reps, etc be in a table.
124 are these hot carcass weights?
134-135 linear effect of sweet bran?? – there were only 2 inclusion rates (thus a guaranteed linear effect)– one in each study. The linear and quadratic effects of DGS are never really “discussed” as such in the results.
131-141 please clarify/specify – were pens used as the experiment unit in the analysis?
146 rephrase – “ … solubles tended to have greater …”
155-156 this statement is misleading – effects for 18, 20 and 30% were not statistically different – so the “P < 0.02” seems inappropriate.
159 and throughout mns. Use “inclusion rate”
Figure 2 include all data points, not just the treatment means.
159-163 these two sentences are repetitive – they say the same thing – just from a different point of view.
163-164 unclear – suggest “ plus solubles ( …) increased gain:feed.”
Table 1 why use SED (which is usually limited to use to compare 2 treatments) in table 1 and SEM in others? Be consistent – preferable with SEM, esp. in Table 1. The number of pens for 15% DG seems to be incorrect – two studies with 6 pens in each were used for these means according to the M&M.
Tables initial weigh is 330 kg in all treatments and tables. Authors need to mention in M&M how this was accomplished and calculated? Were individual pen values for initial BW and other variables somehow mathematically/statistically corrected or adjusted? In table titles or footnote list the experiments that were used in the analysis.
198-199 “…fed EFC processed as DRC … 4.8% (P < ???) increased in G:F …”
200-202 a complex and incomplete sentence. What do you mean “when compared to the individual study”? there was only one study at 25% or 35% SB. Why would differences be statistically different in this analysis but not in the original analysis of the single experiments? You are still only comparing one experiment aren’t you.
205 change “largest” to “larger”. delete “the”. provide P value.
Table 2 title “… diets with 25 or 35% commercial corn gluten feed on finishing “
216, 217, 220 “… with 18 or 20% distillers grains …”
Table 3 include the percent distillers grains in the title.
229-235 & Table 4 these treatments are never mentioned in the M&M.
244 should this be “25 or 35%”
244 and throughout avoid use of “show” or “showed”. Here – “Sweet Bran improved feed efficiency…”. Actually “Commercial corn gluten feed improved feed efficiency”
245 do you really mean “a response” or do you mean “a lack of a response” ?– which is actually what you had.
251-253 this sentence is unclear – I am not sure what treatments you are comparing – DGs or corn hybrid.
269-270 wouldn’t a greater DMI with similar ADG result in lower (poorer) G:F rather than an “improvement in G:F”?
271-273 are you sure this makes for more accurate conclusions – especially with the small sample size in this study ? – without a standard treatment in all studies are we really getting more accurate results or just results that support our bias? Most of the comparisons made in the paper are not really “pooled” – most are still from only one study.
278 & 287 this probably “misses the point”. Steam flaking’s major effect is on ruminal starch digestion – which results in less total starch (g/d) passing to and being digested in the lower gut (compared to DRC) – although the % digestion in the lower gut is probably higher for SFC than DRC.
284-287 awkward, incomplete sentence. Avoid “puzzling“ statement. I’m not sure why you feel these results are puzzling. When the diet contains less starch (i.e. more DGS less corn), then, any effects on grams of starch digested would be smaller. Similarly, the more digestible the starch (ie. SFC or HMC vs DRC) the less effect an added enzyme would have on starch digestion. Total starch digestibility of SFC and HMC is essentially 99%; whereas its closer to 90% for DRC – thus an added enzyme has little chance of “helping” digestion of starch from SFC or HMC but can potentially “help” digestion of DRC.
288 “[8]” is the wrong reference. EFC was not fed in that study. Double check all references.
294 and throughout once an acronym is defined, use it throughout the manuscript (but define in tables and figures)
296 was more efficient than what? Please complete comparison
311 should we really discuss a 1.9% difference – especially with a P value of 0.61? This seems like a real stretch to me.
327 again – but these differences were not statistically different.
332 shouldn’t this be “ but improvements are not expected when…”
Author Response
see attached document

Reviewer 2 Report
I believe completing this analysis of these experiments is information to ensure cattle feeder have the “right” scientific data to make informed decision.
Line 6: extra space before Department
Line 43: Sentence starting with Therefore… You have provided support for this statement.
Line 85: Under Materials and Methods – you have Experiment 1 & 2 from the source reference [10] with the same treatments 1) CON fed as DRC w/ 15% DDGS and 2) EFC fed as DRC w/ 15% DDGS. This appears that you are utilizing the treatments twice. You need to clarify how these are different or remove the treatments.
You need to clean up the replication of data, references [8] and [10] appears to have gone from reference [13]. Something needs to be removed, this reviewer would suggest removing the reference [13]. If there is unique data that is not included in references [8] or [10] publish that data prior to including in these analyses.
Line 95: Sweet Bran – you should include a description of Sweet Bran since this can be different depending on the source.
Line 108: fifteen replications per treatment. It would be helpful to clarify that it is a total of 30 replications per treatments since you have 2 experiments 4 and 5.
Line 142: It would be helpful to include a table with the diet formulation of each of the various treatments. This reviewer would recommend adding description of each diet.
Line 148: Where did you define DG? Sorry if I missed you defining this abbreviation.
Line 153: This reviewer has difficult with the inclusion of numerically greater until you are going to use continuously throughout the paper, which would not be recommended. Please remove.
Line 165: Figure 1 text – P values should be same font size in figure. Figures should be able to stand by themselves, so you should define treatments.
Line 168: Font size should be same for P values. Figures should be able to stand by themselves, so you should define treatments.
Line 183: Table 1 comments: 15DG according to material and methods there should be 12 pens; however, you did use the correct number of pens since Experiment 1 & 2 are reporting the same treatments. Needs to be fixed in text.
It seems odd that all of the Initial BW has the same kg of weight. Upon reviewing the experiments from the various articles, not all initial weights were 330 kg. Please recalculate to ensure these numbers are correct.
Line 200: Sentence starting with “When … Sweet Bran [8]” This reviewer did not follow specifically what data you used to calculate these differences.
Line 205: replace “the largest” with increased
Line 231: replace “smaller” with reduced
Line 244: second 25% should be 35%
Line 258: According to the description in materials and methods, you did not have a study with high moisture corn with 35% Sweet Bran. Experiment 3 had Corn fed as DRC with 35% Sweet Bran; however, no HMC. This need to be fixed. Not sure if Materials and Methods is incorrect or table.
Line 269: Jolly-Breithaupt et al., 2019, guessing this should be [10] and Brinton should be [9]
Line 356: unitalicized J. Anim. Sci.
Line 369: doi does not need to be italicized
Line 381: fix reference 14.
Line 398: hyper between 178 and 184
Line 400: “ent30:” ??? not sure what should be here
Author Response
see attached document

Reviewer 3 Report
Line Comments
1-4 Suggest either “Quantitative Summary of ….” or “Statistical Summary of ….” or Quantitative Review of ….”; is “…. Compared to Control Corn ….” needed in Title? – seems obvious as these were individual “Experiments”; suggest “…. Impact of Feeding Finishing Beef Cattle Syngenta Enogen ….’; shouldn’t Trademark symbol be used here and throughout manuscript with Enogen?;
9-10 Suggest “A pooled statistical analysis of …. was performed to determine the impact ….”;
10 Suggest “…. In beef cattle finishing diets.”
12 is “plus solubles” needed? “improved”
15-16 Suggest “Animal performance was similar for Enogen and conventional hybrids when processed and fed as high-moisture corn.”
17 “improved”
19 “pooled statistical analysis”
20-21 “…. for effects on ….”; “finishing beef cattle”
22 no 0% Sweet Bran in trials?
23 “regression.”
24 “decreased”
25 Suggest “improvement” rather than “better”; increased
27 “…. diets containing sweet bran.”
29 “situation”; improved
32 “beef cattle” rather than “feedlot”?
35 “finishing beef cattle”
38 “improve” rather than “maximize”?
61-77 This seems more detailed of a review of individual studies than need be for an Introduction when the purpose of the paper is a pooled statistical analysis of experiments. I recommend paring this down.
131-141 It is common in meta analyses to weight experiments based on 1/SEM squared, n, P-values or some combination in determining the effect size – was something like this done here and if not then why not?
165-167 as presented it is not totally clear what the various P values are referring to?
168-170 as presented it is not totally clear what the various P values are referring to?
171 has “HCW” been defined previously?
212-223 HMC varies greatly – thus details are needed on the HMC evaluated herein, ie moisture and starch contents, processing, time in storage
Table 1 aren’t these model predicted LS Means? – if so then why not designated as such? why is Initial BW constant -- if constant then why P values?; G:F – is 4 digits to right of decimal customary for this parameter? – personally I find F:G easier to discuss; for a broad based Journal more abbreviations should be defined in footnotes and Tables and Figures should be able to stand alone in that regard
Table 2 see Comments for Table 1; why SEM here and SED in Table 1?; again note constant Initial BW, now with much greater variance and much greater P values than in Table1?
Tables 3-4 see Comments for Tables 1 and 2
236-238 Some perspectives on the beef industry warranted with regard to how common feedlots diets >20% DDGS are relative to 0-20%
271-273 justify this statement about “accuracy” of effect size
275 need something more scientific than “go away”
331 shouldn’t this be “are not”?; also, see comment for L212-223 – therefore I think painting with too broad a brush with very few studies for HMC – need to say something to the effect of “under the conditions of these HMC studies . . . .”
Line Comments
1-4 Suggest either “Quantitative Summary of ….” or “Statistical Summary of ….” or Quantitative Review of ….”; is “…. Compared to Control Corn ….” needed in Title? – seems obvious as these were individual “Experiments”; suggest “…. Impact of Feeding Finishing Beef Cattle Syngenta Enogen ….’; shouldn’t Trademark symbol be used here and throughout manuscript with Enogen?;
9-10 Suggest “A pooled statistical analysis of …. was performed to determine the impact ….”;
10 Suggest “…. In beef cattle finishing diets.”
12 is “plus solubles” needed? “improved”
15-16 Suggest “Animal performance was similar for Enogen and conventional hybrids when processed and fed as high-moisture corn.”
17 “improved”
19 “pooled statistical analysis”
20-21 “…. for effects on ….”; “finishing beef cattle”
22 no 0% Sweet Bran in trials?
23 “regression.”
24 “decreased”
25 Suggest “improvement” rather than “better”; increased
27 “…. diets containing sweet bran.”
29 “situation”; improved
32 “beef cattle” rather than “feedlot”?
35 “finishing beef cattle”
38 “improve” rather than “maximize”?
61-77 This seems more detailed of a review of individual studies than need be for an Introduction when the purpose of the paper is a pooled statistical analysis of experiments. I recommend paring this down.
131-141 It is common in meta analyses to weight experiments based on 1/SEM squared, n, P-values or some combination in determining the effect size – was something like this done here and if not then why not?
165-167 as presented it is not totally clear what the various P values are referring to?
168-170 as presented it is not totally clear what the various P values are referring to?
171 has “HCW” been defined previously?
212-223 HMC varies greatly – thus details are needed on the HMC evaluated herein, ie moisture and starch contents, processing, time in storage
Table 1 aren’t these model predicted LS Means? – if so then why not designated as such? why is Initial BW constant -- if constant then why P values?; G:F – is 4 digits to right of decimal customary for this parameter? – personally I find F:G easier to discuss; for a broad based Journal more abbreviations should be defined in footnotes and Tables and Figures should be able to stand alone in that regard
Table 2 see Comments for Table 1; why SEM here and SED in Table 1?; again note constant Initial BW, now with much greater variance and much greater P values than in Table1?
Tables 3-4 see Comments for Tables 1 and 2
236-238 Some perspectives on the beef industry warranted with regard to how common feedlots diets >20% DDGS are relative to 0-20%
271-273 justify this statement about “accuracy” of effect size
275 need something more scientific than “go away”
331 shouldn’t this be “are not”?; also, see comment for L212-223 – therefore I think painting with too broad a brush with very few studies for HMC – need to say something to the effect of “under the conditions of these HMC studies . . . .”
Author Response
see attached document

Round 2
Reviewer 1 Report
To Authors;
There are some errors in the text (or the tables?) that must be corrected.
Also some rewording would improve the clarity and readability of the paper.
I believe the authors still need to be consistent in use of “feed efficiency” or G:F ( which is actually gain efficiency). Be consistent throughout and use only one term.
The manuscript still needs some rewriting.
Line Comment
21 “ when fed either dry rolled (DRC) or high-moisture corn (HMC) …”
27 “… from a 4.8% improvement to …”
34 delete “feeds”
76 define acronym – or better delete it since it is never used.
69 here and throughout – change “observed” to “had”
64-66 vs 68-70 these two sentences contradict each other – both refer to reference 8 but in first sentences G:F is “improved” and in the second is not “(P > 0.17).”
159 and throughout mns. I still believe use of “inclusion rate” is more precise and clearer than “inclusion”
Figure 1 Still believe all data points, not just the treatment means should be in figure.
155-156 All the data presented are estimates/calculated/predicted values – can you provide more specifics on how they were calculated?
192 but the P value is 0.49 in Table 2.
194 but the P values is 0.65 in Table 2.
197 “… a greater 12th rib fat thickness…” . Throughout discussion – 12th rib fat thickness can be considered a surrogate for the quantity or amount of fat – but it is actually just the “thickness” not the quantity or amount of fat. Therefore wording needs to be modified in several locations (line 207, 330)
199 but P value is 0.83 in table 2
201 but P value is 1.0 in table 2.
223,333, et al statistics detect “differences” not “similarity” – therefore the more proper term for “similar” in this and several other cases is “not significantly different.”
303 Still respectively disagree with authors and this statement. I agree steam flaking increases the digestion of starch entering the small intestine (about 93% for SFC vs 72% for DRC: Owens et al 2006); however (assuming 10 kg DMI and diet is 50% starch) it decreases the total quantity of starch digested post-ruminally by almost 50% in finishing cattle (about 750 g/d vs 1,300 g/d for SFC and DRC, respectively) because of its effect on ruminal starch digestion (about 84% vs 64% for SFC and DRC, respectively) – which results in less total starch passing to (about 800 vs 1810 g/d, respectively) and being digested in the lower gut (compared to DRC). Assuming a 10% lower DMI for SFC fed cattle makes the difference in post-ruminal starch digestion even larger (700 vs 1300 g/d, respectively). I believe Huntington suggested there might be a limitation on the grams of starch that can be digested in the lower gut (which Owens vehemently disagrees with BTW) – presumably because of inadequate enzyme secretion. Thus – supplemental amylase might be beneficial with DRC diets but totally unnecessary with SFC or HMC based diets because of limited amylase in the small intestine.
305 etc why not include the papers of Buttrey et al (2012,2013) and work at Texas Tech etc that also supports the corn processing/WDGS discussion?
330 change “amount” to “thickness”
Reviewer 2 Report
Additional information in paper improved article which is valuable to other researchers and educators.
Author Response
Thankyou. No changes suggested
Reviewer 3 Report
HMC varies greatly – thus details are needed on the HMC evaluated herein, ie moisture and starch contents, processing, time in storage
Added in methods
where specifically was this done?
Tables
aren’t these model predicted LS Means? – if so then why not designated as such?
where specifically was this done?
how can Initial BW be 330 kg with negligible variance across studies and among treatments?
It is common in meta analyses to weight experiments based on 1/SEM squared, n, P-values or some combination in determining the effect size – was something like this done here and if not then why not?
No, as we have the pen means, so not a meta-analysis but a pooled analysis using meta-analysis techniques. We have been debating how best to describe this as most understand meta-analysis without pen means.
why can't a meta analysis be done using treatment means and variance estimates when Pen was used as the experimental unit in the individual studies?
Round 3
Reviewer 1 Report
no additional suggestions.
Reviewer 3 Report
Thanks for considering and addressing concerns. Recommended acceptance.